# Portable Mid-Infrared Spectroscopy Combined with Chemometrics to Diagnose Fibromyalgia and Other Rheumatologic Syndromes Using Rapid Volumetric Absorptive Microsampling

**DOI:** 10.3390/molecules29020413

**Published:** 2024-01-15

**Authors:** Shreya Madhav Nuguri, Kevin V. Hackshaw, Silvia de Lamo Castellvi, Haona Bao, Siyu Yao, Rija Aziz, Scott Selinger, Zhanna Mikulik, Lianbo Yu, Michelle M. Osuna-Diaz, Katherine R. Sebastian, M. Monica Giusti, Luis Rodriguez-Saona

**Affiliations:** 1Department of Food Science and Technology, The Ohio State University, Columbus, OH 43210, USA; nuguri.2@buckeyemail.osu.edu (S.M.N.); delamocastellvi.1@osu.edu (S.d.L.C.); bao.172@buckeyemail.osu.edu (H.B.); giusti.6@osu.edu (M.M.G.); rodriguez-saona.1@osu.edu (L.R.-S.); 2Department of Internal Medicine, Division of Rheumatology, Dell Medical School, The University of Texas, 1601 Trinity St., Austin, TX 78712, USA; 3Campus Sescelades, Departament d’Enginyeria Química, Universitat Rovira i Virgili, Av. Països Catalans 26, 43007 Tarragona, Spain; 4Department of Nutrition and Food Hygiene, School of Public Health, Southeast University, Nanjing 210019, China; siyuyao@seu.edu.cn; 5Department of Internal Medicine, Dell Medical School, The University of Texas, 1601 Trinity St., Austin, TX 78712, USA; rija.aziz@austin.utexas.edu (R.A.); scott.selinger@austin.utexas.edu (S.S.); michelle.osuna@austin.utexas.edu (M.M.O.-D.); kate.sebastian@austin.utexas.edu (K.R.S.); 6Department of Internal Medicine, Division of Immunology and Rheumatology, The Ohio State University, 480 Medical Center Dr, Columbus, OH 43210, USA; zhanna.mikulik@osumc.edu; 7Center of Biostatistics and Bioinformatics, The Ohio State University, Columbus, OH 43210, USA; lianbo.yu@osumc.edu

**Keywords:** fibromyalgia, rheumatic diseases, central sensitization, FT-MIR, OPLS-DA, point-of-care device

## Abstract

The diagnostic criteria for fibromyalgia (FM) have relied heavily on subjective reports of experienced symptoms coupled with examination-based evidence of diffuse tenderness due to the lack of reliable biomarkers. Rheumatic disorders that are common causes of chronic pain such as rheumatoid arthritis, systemic lupus erythematosus, osteoarthritis, and chronic low back pain are frequently found to be comorbid with FM. As a result, this can make the diagnosis of FM more challenging. We aim to develop a reliable classification algorithm using unique spectral profiles of portable FT-MIR that can be used as a real-time point-of-care device for the screening of FM. A novel volumetric absorptive microsampling (VAMS) technique ensured sample volume accuracies and minimized the variation introduced due to hematocrit-based bias. Blood samples from 337 subjects with different disorders (179 FM, 158 non-FM) collected with VAMS were analyzed. A semi-permeable membrane filtration approach was used to extract the blood samples, and spectral data were collected using a portable FT-MIR spectrometer. The OPLS-DA algorithm enabled the classification of the spectra into their corresponding classes with 84% accuracy, 83% sensitivity, and 85% specificity. The OPLS-DA regression plot indicated that spectral regions associated with amide bands and amino acids were responsible for discrimination patterns and can be potentially used as spectral biomarkers to differentiate FM and other rheumatic diseases.

## 1. Introduction

Fibromyalgia (FM) is a chronic musculoskeletal pain syndrome characterized by widespread body tenderness as a result of aberrant responses to direct pressure, temperature and a variety of other sensory stimuli [1]. In addition to pain, affected individuals frequently manifest multiple other symptoms of central sensitization, including mood and sleep disturbances, intestinal irritability, migraine, temporomandibular joint disorder, and chronic fatigue, amongst others [1,2,3,4,5]. FM appears to result from the dysregulation of the central nervous system (CNS) and neuroendocrine functioning leading to central sensitization where the neuronal signals are amplified, causing greater pain sensation [3]. The prevalence is nearly 5% of the population globally with at least a 5-fold predilection for women [6]. In childhood, early signs of fibromyalgia may be manifested by “growing pains” or childhood migraines. Typically, adults present for evaluation aged between 30 and 40 years; however, all ages are affected [7]. FM has a substantial impact on the economy, with healthcare expenses being substantially higher than those of unaffected individuals and comparable to the costs of care associated with rheumatoid arthritis [6].

The American College of Rheumatology criteria for FM have evolved from a dependence on tender points to consideration of a widespread pain index (WPI) coupled with a symptom severity scale score (SSS) [2]. Rheumatic diseases such as chronic low back pain (CLBP), rheumatoid arthritis (RA), systemic lupus erythematosus (SLE), and osteoarthritis (OA) may frequently have symptoms of comorbid FM, which makes the disorder an even more challenging condition to identify [3], often taking nearly 5 years to reach a definitive diagnosis [8].

Vibrational spectroscopy (infrared and Raman) captures characteristic chemical “fingerprints” of the molecules, including biomarkers present in the biological samples. This technology offers a high-throughput and rapid diagnosis of diseases with minimum sample preparation and is being applied in several areas of the medical field such as cancer [9,10], rheumatology [2], and urology [11]. Fourier-transformed infrared (FT-MIR) spectroscopy acquires the unique biochemical signals by irradiating a polychromatic light source (4000 to 650 cm^−1^) on the sample and recording the absorbance by the vibrating functional groups of the global metabolites in the sample. Spectral data containing a plethora of information undergo extensive analysis using multivariate pattern recognition techniques. These methods identify the key biological components of each disease and enable the development of algorithms that can be deployed in IR instruments at the point of care. Soft independent modelling of class analogy (SIMCA) and orthogonal partial least squares discriminant analysis (OPLS-DA) are examples of multivariate supervised classification algorithms. In supervised learning, the disease or health status corresponding to each spectrum is provided. The methodology builds an algorithm by critically assessing the differences in absorption frequencies among the different groups potentially arising from the disease biomarkers.

Our group has been extensively evaluating the spectral features of FM and other rheumatologic disorders over the past decade. Initially, we assessed the capabilities of mid-IR microspectroscopy to differentiate FM from OA and RA; the clusters were accurately differentiated using SIMCA analysis [12]. Subsequently, these samples were subjected to metabolomic analysis using LC-MS/MS which revealed the similarities between the OA and RA groups, distinct from FM subjects. Later, we assessed the feasibility of portable MIR and FT-Raman microspectroscopy for differentiating FM patients and additionally performed uHPLC-PDA-MS/MS analysis to complement our insights from vibrational techniques [2]. Functional groups corresponding to peptide backbones and pyridine-carboxylic acids were identified as crucial in discriminating FM from other central sensitivity syndromes. We further examined the effect of different extraction procedures on the performance of the OPLS-DA classification algorithm [13]. These procedures included the evaluation of pure blood samples, protein precipitation using organic solvent (acetonitrile), and physical extraction involving semi-permeable filter membranes. Comparatively, the chemical precipitation and washed-membrane-based extraction exhibited a higher performance than direct measurements of blood aliquots. Recently, our group has used the portable MIR for the metabolic phenotyping of the spectral signatures to diagnose clinically similar long COVID and FM specimens [14]. Interestingly, through deconvolution of the pre-processed spectra, we identified a distinct spectral marker (1565 cm^−1^) in FM samples contrasting with long COVID. This spectral marker is primarily associated with the side group of glutamate amino acids. OPLS-DA classification analysis based on the spectral region 1500–1700 cm^−1^ resulted in a perfect classification with 100% accuracy. Overall, we have made significant progress in vibrational spectroscopy-based metabolomics for diagnosing the FM disorder.

We used dried bloodspot (DBS) cards in our previous studies due to reduced blood volume, easy sample collection, easy transport, and storage feasibility [15]. The Whatman filter cards have pre-printed circles which serve as a reference while sampling the blood on the paper. The viscosity of the blood sample is greatly influenced by hematocrit levels that define the concentration of red blood cells in a unit volume. Samples with varying hematocrits can exhibit differences in diffusion rates when applied to filter cards, resulting in heterogenous volumes within the pre-defined circles [16]. The Neoteryx Mitra ^®^ (Torrance, CA, USA) device is based on a novel volumetric absorptive microsampling (VAMS) approach for blood sampling. In VAMS technology, a fixed volume of blood is absorbed onto a porous substrate; the volume absorbed is dependent on the characteristics and concentration of the substrate [17]. The Mitra sampler device comprises two plastic spindles, each with an adsorbent VAMS tip that wicks a fixed volume of blood [16]. The sampler is then placed in a protective plastic clamshell and locked in specimen bag with desiccant [18]. The VAMS-based microsampling approach offers several advantages over DBS, including reduced sample volume requirements, easy transportation and storage, and controlled sample volume bias [16,19,20].

The objective of the current study was to develop a diagnostic approach using the characteristic infrared signatures of blood samples collected with the VAMS technique for the real-time diagnosis of the FM disorder from other rheumatological diseases (RA, SLE, OA, CLBP). The incorporation of the VAMS technique aims to maintain a consistent blood volume across all samples.

## 2. Results

### 2.1. Clinical Characteristics of Subjects

The clinical characteristics of the patients with FM, non-FM, and NCs are presented in Table 1. Subjects with FM (*n* = 179; F:164, M:15) had a mean age of 43.1 ± 13.4 with a range of 18–73. Their BMI was 32.0 ± 9.2, with a mean FIQR of 48.2 ± 26.0. The CSI was 58.1 ± 22.9 with an MPI of 90.5 ± 51.7 and a BDI of 19.3 ± 11.8. Patients with rheumatic diseases without FM (non-FM) (*n* = 158; F: 122, M: 36) had a mean age of 50.5 ± 15.98 with a range of 18 ± 81. Their BMI was 28.2 ± 11.3, with a mean SIQR of 33.5 ± 23.0. The CSI was 27.8 ± 21.9 with an MPI of 41.7 ± 29.6 and a BDI of 9.5 ± 8.6. Thirteen control samples (NCs) were utilized as a further comparator. These subjects were on no medications and did not have FM or any underlying rheumatic disorder, including RA, SLE, OA, or CLBP. Table 2 shows the statistically significant differences seen between the FM and non-FM groups with respect to the CSI, SIQR/FIQR, MPQ, and BDI values.

### 2.2. Mid-Infrared Spectroscopy

Figure 1a shows a representative spectrum of extracted LMFs from FM, other rheumatic diseases (SLE, OA, CLBP, RA), and NC patients. The absorbance intensities and spectral profile of each disease and NC samples were similar. The broad absorption band around 3600–3100 cm^−1^ was associated with O-H stretching vibrations and overtones of the amide II bond (N-H in-plane bending and C-N stretching) [21]. The absorption bands at 2800–3000 cm^−1^ were linked to C-H stretching vibrations in metabolites. The less intense absorption bands at 2960 and 2920 cm^−1^ can be linked to the asymmetric stretching of sp^3^ and sp^2^ hybridized C-H groups, respectively, while the symmetric methyl and methylene C-H stretching vibrations can be related to 2850 cm^−1^ [22,23]. The infrared region from 1500 to 700 cm^−1^ is known as the fingerprinting region. The absorption band at 1740 cm^−1^ can be attributed to C=O stretching, mainly from aldehydes or ester linkages present in lipids and cholesterol [24,25]. The most intense absorption band at 1580 cm^−1^ accompanied by a weak shoulder at 1660 cm^−1^ can be linked to the conformational changes in the amide I and II groups (C=O stretching) within the peptide backbone and the nitrogenous bases of DNA involving C=O and C=N bonds [13,24]. The presence of the absorption band around 1400 cm^−1^ is influenced by C-H umbrella deformations [26] and the hyperconjugation effect on methyl bending modes [25]. Furthermore, vibrations associated with the phosphodiester bonds in DNA, RNA, and phospholipids contributed to the bands at 1245 cm^−1^ and 1088 cm^−1^ [25]. Additionally, bands around 1000 cm^−1^ can be credited to the stretching of C-O-C in cholesterol, phospholipids, and triglycerides [25] and the stretching of C-O in nucleic acids and polysaccharides [24]. Finally, it has been observed that the region around 985–1000 cm^−1^ is indicative of Ig protein characteristics [24,27].

The second derivative transformation was performed using a Savitzky–Golay polynomial filter [28]. In this method, consecutive subsets of the 19-point-sized windows were fitted by a second-order polynomial, followed by the application of a second derivative to the fitted polynomial curve [29]. Figure 1b compares the transformation (normalized and second-derivatized with 19 points) applied to the averaged spectra of FM, non-FM, and NC subjects.

Figure 2 shows the spectral signature of an empty VAMS (blank) extracted under the same conditions as the samples to recognize the contribution of this microsampling support to the spectral signatures of our samples. The spectrum of the blank, when compared with the intensities of the sample, revealed that the processing and storage factors contributed negligibly to the absorbance. This ensures that the spectral signatures of the sample are unique to the sample and are not influenced by external artifacts.

### 2.3. Classification Analysis

Based on the spectral data, the differences among FM and other rheumatic disease signals were visualized through statistical analysis of OPLS-DA. The OPLS-DA algorithm developed using the 2600–3680 cm^−1^ and fingerprint (1740–785 cm^−1^) regions gave the best performance. The raw spectra were pre-processed (mean-centered, normalized), and second derivative (SG, 19 points) was applied with one orthogonal signal correction (OSC) component, which made the predictive quality of the model satisfactory [30]. The regression vector plot (Figure 3a) demonstrated that the discrimination was dominated by 1590 and 1562 cm^−1^ followed by 1411 cm^−1^, 1391 cm^−1^, 1024 cm^−1^, and 1074 cm^−1^. The information at 1590 and 1562 cm^−1^ can be attributed to amide bands and amino acids. The bands in the region 1430–1360 cm^−1^ have been linked to ν(COO^−^) from amino acids, while the region around 1024 cm^−1^ is associated with contributions from ν(C=O) and ν(C-O), particularly originating from the ribose unit of RNA [24]. Furthermore, the absorption at 1074 cm^−1^ may be attributed to Ig proteins and ν_s_ (PO_2_^−^) of DNA, RNA, and phospholipids [24].

Six model factors were selected (Figure 3c) that cumulatively described 81.4% of the variation, with an excellent R^2^ of 0.99 and a low SECV (0.02). An appropriate model complexity that levels between underfitting and overfitting is required for accurate predictions. Insufficient descriptors may not capture enough variance that explains the data, leading to underfitting, while a higher number of factors can introduce noise, causing overfitting of the model [31,32]. Moreover, various regression objects, such as the regression plot, outliers’ diagnostics, and performance metrics, are dependent on these factors.

The explained variation in the X matrix by the factors (Figure 3b) did not follow a gradual decrease, as seen in the principal component analysis (PCA). This deviation is explained by the fact that OPLS-DA emphasizes the relationship between the predictive variables in X and the Y response [33,34]. The OSC filter selectively removes the systematic X variations that are not response-predictive, focusing on factors that maximize the discrimination among the classes [33]. This contrasts with PCA, which captures and maximizes the overall variation within the X matrix [34]. Nevertheless, with the inclusion of more factors, the associated SECV decreased (Figure 3c).

Less than 11% of the data points were found to be influential with a high leverage and studentized residual; they were eliminated to improve the predictive accuracy of the model. Figure 3b illustrates the score plot distinguishing FM subjects from other rheumatic disease in the calibration dataset. The score plot represents the sample coordinates projected in a 3D hyperspace defined by the first three factors [35].

The OPLS-DA classification method incorporates the regression proficiencies of the PLS approach, where the response matrix is quantitative. The Y matrix included binary labels 0 and 1, indicating non-FM and FM cases, respectively. A leave-one-out cross validation approach was used to develop the calibration model in which each sample from the training set was temporarily left out and the remaining samples were used to build the model to predict the left-out sample. The cumulative Y residuals resulting from this internal validation were recorded as SECV. The SEP is based on external validation and represents the residuals of the test set predictions and their corresponding known class labels. The SECV and SEP were similar, with a low value of 0.02 (Table 3).

The calibration OPLS-DA model produced a distinct cluster for the FM and non-FM categories with zero misclassifications, resulting in 100% specificity, 96% sensitivity, and 98% accuracy (Table 3). Sensitivity assesses the model’s ability to accurately identify the cases of FM in a population with FM disease, while specificity estimates the proportion of correctly identified non-FM cases within a total group of subjects without FM disease [36]. The accuracy is a global measure that calculates the proportion of correctly classified subjects in a population having all the diseases considered in the training model [36]. OPLS-DA has powerful modeling features that facilitate separation based on experimental classes in the calibration model, yielding positive results [33]. An external validation with an unseen dataset is required to validate the statistical significance of the separation and draw conclusions [33]. When validating with an unseen test set, the model showed good results with 83% sensitivity, 85% specificity, and 84% accuracy (Table 3). Thus, our algorithm will correctly predict an unknown subject as FM or non-FM with an 84% accuracy rate.

The ROC plots the sensitivity and specificity of the model’s response to all possible thresholds that define the result as positive [37]. Additionally, it summarizes the predictive accuracy of the models by providing the AUC; larger areas having values closer to 1 signify a higher accuracy. The diagnostic test achieved an AUC of 0.88, categorizing its accuracy as “good” [38]. The 95% confidence interval of the AUC computed using the DeLong method ranged from 0.79 to 0.96 (Figure 4).

The generated calibration model was used to test 13 NC samples. An outlier was observed in the considered NC samples and eliminated. The model successfully classified the remaining 12 NCs as non-FM, facilitating its application to differentiate FM patients from NCs.

## 3. Discussion

The findings of this study are intriguing. The clinical groups were generally similar in terms of age and BMI. The FIQR is a validated surrogate marker of pain in subjects with FM. Similarly, the SIQR is a FM-neutral questionnaire which asks identical questions but does not assume that patients have FM. Patients with rheumatic conditions and no comorbid FM had statistically significant differences in terms of lower SIQR, BDI, MPQ, and CSI values relative to patients with FM, which is expected given the nature of these disorders. The FM group was 91.7% female, while the non-FM group was 77% female. This may slightly limit our ability regarding the generalizability of our data due to minor gender differences, although these variances are not profound. A further confounder could also be due to differences in medication usage between the two groups. Medications of the recruited patients were recorded at the time of blood collection. This study was not powered to determine the effect of medication on spectral profiles, although this is a goal of our group in the future. Future studies will seek to mitigate a possible medication effect in the analysis. We can record patient medication usage, categorize the medication by types, and quantify it by amount usage. Furthermore, we can associate the medication usage with each metabolite biomarker with correlations or logistic regression models. Finally, we can compare subjects on similar medications while eliminating confounder medications from the analyses.

In this study, we evaluated the application of a portable FT-MIR method combined with chemometrics for the rapid diagnosis of the FM disorder from other rheumatological diseases (OA, RA, SLE, CLBP) on samples collected using VAMS technology. Our lab has been extensively evaluating the spectral signatures delineating features unique to FM metabolites [2,12,13,14]. Initially, we investigated the ability of MIR microspectroscopy to identify FM patients from RA and OA. SIMCA classification analysis differentiated the class of interest with no false predictions [12]. Subsequently, we evaluated the feasibility of a portable FTMIR and Raman microspectroscopy in identifying FM cohorts (*n* = 50) from other rheumatologic patients (RA = 29, OA = 19, SLE = 23) using SIMCA pattern recognition analysis. Following this, our group investigated the effect of various extraction protocols, including direct blood measurements, physical extraction using filter membranes (unwashed and rinsed with water to eliminate the artifact, glycerol), and chemical extraction using acetonitrile, on the predictive algorithms of FM (*n* = 122) from other central sensitivity syndromes (RA = 43, SLE = 17, OA = 10) [13]. With the increased number of samples, the non-predictive variation among the classes also increased. OPLS-DA was employed to remove these orthogonal variables. The statistical results of the OPLS-DA models revealed a higher performance for washed-membrane- and chemical-based extraction protocols when compared with direct blood measurements and unwashed semi-permeable filters. Earlier this year, we made a similar approach to diagnosing clinically similar long COVID and FM patients using the unique metabolic signatures of low-weight molecules extracted using washed filter membranes [14]. The OPLS-DA algorithm based on the amide region successfully classified the cases into their respective diseases.

The previous studies incorporated the conventional blood cards for sampling and storing the blood aliquots [2,12,13,14]. In the current study, we increased the sample size (*n* = 337) with approximately balanced representation in each cluster (FM = 179 and non-FM = 158). A larger sample size incorporates variations that can be expected in real clinical scenarios, contributing to the development of a more reliable model. Samples were acquired using a novel VAMS technique presented by Neoteryx Mitra^®^ that ensured consistent blood volume collection controlling the variations in sample quantities. The OPLS-DA algorithm showed an excellent performance, with a high R^2^ and low error (SECV and SEP) signifying a good discrimination of FM specimens from other rheumatic diseases. Infrared vibrations of amide I and amino acids dominated the separation, followed by aromatic compounds and nucleic acid functional groups. The results were in agreement with our previous studies, which also highlighted the protein backbone and pyridine-carboxylic acid region to be important in class separation [2,13,14]. These functional groups may possibly be linked to pain-processing neuropeptides and neurotransmitters, principally glutamate and tryptophan (serotonin precursor), whose levels have been found to be different in the FM population [1,39,40,41]. The contribution of aromatic metabolites towards enhanced model performance encourages the exploration of Raman spectroscopy and surface-enhanced Raman spectroscopy (SERS), given their superior accuracies in detecting aromatics and alkenes [42]. Currently, our group is working on surface-enhanced Raman spectroscopy (SERS) to identify the trace characteristic metabolites of FM and other related syndromes. External validation of the generated classification model showed “good” performance with a sensitivity, specificity, and accuracy ranging from 83% to 85%. Considering the ROC plot, an AUC of 0.88 was computed with a 95% confidence interval from 0.79 to 0.96.

The ambiguity of FM symptoms with other rheumatological disorders makes it difficult for physicians to accurately diagnose the disease. While some physicians try to follow the updated diagnosis criteria, others may believe the disorder is “psychological”, discouraging patients from pursuing any treatments [2]. Despite adhering to the revised criteria for FM identification, the specimens are often incorrectly diagnosed due to significant errors associated with high levels of subjectivity in the survey forms [2,43]. Given that no reproducible biomarker exists for the FM disease, both patients and physicians are keenly seeking an objective assessment and a gold standard for defining and detecting FM [2]. Further, the current evaluation takes a very long time (approximately 5 years) to receive a reliable result, leading to improper and unnecessary treatments that may worsen the symptoms. Our studies have been contributing towards developing a rapid and reliable diagnostic test and identifying possible biomarkers or therapeutic targets for advancing treatment approaches. This study entails the application of a novel VAMS technique that considers a homogeneous sample collection and reduces the variations not associated with the disease, facilitating the important variables to be investigated by the model, improving its performance.

## 4. Materials and Methods

### 4.1. Patient Recruitment and Blood Sampling

Approval from the University of Texas at Austin institutional review board was obtained prior to embarking on any human subject studies. All studies adhere to the Declaration of Helsinki principles. The IRB approval date was (study no. 2020030008) 19 June 2020. Following informed consent, blood samples were obtained from patients with FM (*n* = 179) and other rheumatic disorders (RA, SLE, OA, CLBP) (non-FM (*n* = 158)) and healthy controls (*n* = 13) at University of Texas at Austin clinics located at University Texas Health Austin Clinics, Austin, Texas, and The Ohio State University Rheumatology Clinics located at Carepoint East, Columbus, Ohio, between September 2020 and June 2023. Blood samples were collected intravenously and stored on Neoteryx Mitra devices (Neoteryx, CA, USA) with a 30 µL total collection volume tip employing VAMS technology. Each device was equipped with two tubes sampled from the same patient, providing analytical replicates. The samples were shipped to the Rodriguez-Saona’s Vibrational Spectroscopy laboratory at The Ohio State University and kept at −20 °C until extraction.

All subjects provided self-reports of symptoms through use of the Revised Fibromyalgia Impact Questionnaire Revised (FIQR), a 10-item self-rating instrument that measures physical functioning, work status, depression, anxiety, sleep, pain, stiffness, fatigue, and wellbeing [44]. The Beck Depression Inventory (BDI) is a 21-item, self-report rating inventory that measures characteristic attitudes and symptoms of depression [45]. The Symptom Impact Questionnaire Revised (SIQR) is the FM-neutral version of the FIQR and does not assume that the patient has FM [46]. The SIQR was utilized as a measure of physical functioning, work status, depression, anxiety, sleep, pain, stiffness, fatigue, and wellbeing for all subjects without FM and normal controls. The Central Sensitization Inventory (CSI) is a two-part patient-reported outcome measure that assesses somatic and emotional symptoms common to CSS [47]. The McGill Pain Questionnaire (MPQ) is an instrument providing descriptive aspects of pain as well as pain intensity.

Criteria for the diagnosis of FM included age 18–80 with a history of FM and meeting current criteria for FM [43,48,49]. Criteria for diagnosis of osteoarthritis (OA) inclusion involve subjects aged 18–80 with morning stiffness < 30 min in duration, crepitus, and radiographic evidence of OA or clinician confirmation with a lack of evidence of a concurrent inflammatory component. Chronic low back pain subjects’ inclusion criteria were age 18–80 with low back pain for at least 3 months and meeting the criteria of the American Pain Society [50]. Systemic lupus erythematosus inclusion criteria involve subjects that are aged 18–80 with defined SLE according to the revised ACR classification criteria [51]. Rheumatoid arthritis (RA) inclusion criteria were age 18–80 and meeting ACR criteria for rheumatoid arthritis [50,51,52]. Sigmaplot v15.0 and SigmaStat v4.0 software (Inpixon, Palo Alto, CA, USA) were utilized for statistical analysis of questionnaires.

### 4.2. Sample Extraction

Low-molecular-weight solutes (LM) were isolated from high-molecular fractions in the sample using Amicon ultra centrifugal filter units (Sigma-Aldrich, Inc., St. Louis, MO, USA). The optimized washed-filter-based protocol from our previous studies was used to extract the serum fractions [13,14]. Prior to extraction, the filter tubes were rinsed with 3 mL of Milli-Q water followed by centrifugation (Legend™ XFR Centrifuge, Thermo Scientific, Waltham, MA, USA) at 4000 rpm for 10 min (Figure 5a). The filters were washed four times to remove the glycerol present on the filter that creates artifacts in the IR spectral data and may interfere with the chemometric analysis.

The Neoteryx tips (Neoteryx, CA, USA) containing blood aliquots were placed in 2 mL Milli-Q water and sonicated for 15 min to efficiently extract blood from the tip. Then, the solutions containing blood samples were transferred into the washed filters and centrifuged at 4000 rpm for 15 min at 40 °C. Low-molecular fractions permeated through the membranes while high-molecular solutes remained on the filter. Overall, the centrifugal filter membranes (10 MWCO KDa) purified LM fractions (LMFs) containing important water-soluble metabolites for plasma-based identification of biomarkers [53]. The LMF permeate solutes were nitrogen-flushed (BT Lab System, St. Louis, MO, USA), followed by vacuum drying (Eppendorf Vacufuge plus, Eppendorf, Hamburg, Germany) to remove water and obtain a plasma film (Figure 5b). These films were stored at −20 °C until the IR analysis. Additionally, an analysis was conducted on a blank VAMS substrate to assess the contribution of the processing factors to the IR data.

### 4.3. Infrared Spectroscopy Analysis

Mid-infrared analysis was performed using a portable Agilent 4500 series (Agilent Technologies, Santa Clara, CA, USA) Fourier-transformed mid-infrared instrument, equipped with a triple-bounce diamond attenuated total reflectance (ATR), covering a spectral range from 4000 to 700 cm^−1^. The instrument featured a 200 µm active area on a 2 mm diameter sampling surface, offering an estimated penetration depth of 6 µm at 17 cm^−1^. The unit was outfitted with a zinc selenide beam splitter, a high-throughput Michelson interferometer, and a thermoelectrically cooled deuterated triglycine sulfate (dTGS) detector [54]. To prepare the samples for spectral collection using the portable FTIR, LMFs were reconstituted with 6 µL of Milli-Q water and vortexed for 10 s. Subsequently, 1 µL of this solution was placed onto the ATR and vacuumed to create a film on the spectral window. The sampling area was cleaned with 70% ethanol, and a background spectrum was acquired before each reading. A 4 cm^−1^ resolution was employed with co-addition of 64 scans to improve the signal-to-noise ratio.

### 4.4. Pattern Recognition Analysis

Pattern recognition software, Pirouette (Pirouette version 4.5, Infometrix Inc., Woodville, WA, USA), was employed to resolve the spectral information relevant to each class (FM vs. RA, SLE, OA, CLBP) and identify spectral differences among them using multivariate statistical methods. Classification algorithm OPLS-DA was used to predict FM samples from other musculoskeletal disorders. OPLS-DA augments the supervised regression technique, PLS, by incorporating an orthogonal signal correction (OSC) filter. With reference to the response variable (Y), the spectral information (X) matrix is structured into variations that are predictive and non-predictive (orthogonal) to Y [55]. The OSC identifies the Y-orthogonal variation and removes or retains it based on its importance in describing the class separation. In general, the Y-orthogonal vectors explain within-class variance and are eliminated, facilitating the interpretation of important predictive variables by the algorithm [55]. The dataset was divided into training and validation sets using the “Random” algorithm [30] (Table 4). In total, 80% of the dataset was used to build the calibration model (*n* = 179 FM and *n* = 158 other diseases), and the generated model was used to externally validate the remaining 20% of the dataset (*n* = 35 FM and *n* = 27 other diseases).

For the multivariate analysis, the dataset was mean-centered, and pre-processing methods, namely, normalization and second derivative (window size 19), were used to optimize the performance of the model. Mean centering creates a zero-mean matrix, making it convenient to compare inter-variable relationships instead of using absolute values [30,44]. The second derivative resolves overlapping information, identifying peaks that explain the classification of groups while also eliminating baseline errors. Smoothing enhances the chemical signals in the spectra by reducing the high-frequency noise [45], while normalization scales the peak intensities in the spectra to a value of 100. Important spectral regions discriminating FM samples from other disorders were selectively considered through the elimination of noisy wavenumbers and signals not explaining the differences among the groups. This independent variable selection has been found to improve the statistical performance [46,56]. The number of latent variables corresponding to a low variance was chosen to avoid overfitting by considering a smaller number of factors where no noticeable improvement in the performance was observed with further inclusion of latent variables. Outliers were identified based on leverage and the studentized residual. The studentized residual was considered for determining outliers to account for the leverage. A critical boundary for detecting outliers was defined by the Pirouette software based on the Mahalanobis distance [30]. Data points exceeding these critical values were assessed carefully. The samples with a high difference from the training set’s profile had considerable control in the model and were excluded from the analysis.

An internal validation, the leave-one-out approach, was used during model development, where one of the training samples is excluded and predicted by the model built using the other samples in the training set. This is iterated until all the samples have been removed and computed. The internal cross validation provided a performance estimate of the calibration model, including the correlation coefficient of cross validation (R^2^) and standard error of cross validation (SECV), while external validation with an unseen dataset revealed the standard error of prediction (SEP) and presented the model’s performance when deployed in real-world scenarios for FM diagnosis. Furthermore, the statistics of internal and external validation described the performance of the diagnostic model through sensitivity, specificity, and accuracy. In addition, a receiver operating characteristic (ROC) plot was generated in R [57] using the pROC package, v1.18.5 [58] based on the results of external validation. The plot provided the area under the curve (AUC) with a 95% confidence interval, which assessed the accuracy of the diagnostic model. The AUC was computed using the DeLong method [59,60].

## 5. Conclusions

In this study, we evaluated the application of a portable FT-MIR instrument for the inexpensive and rapid diagnosis of the FM disorder from other central sensitivity syndromes. Low-molecular metabolites were extracted using an optimized sample preparation protocol that incorporates washed membranes to filter the blood. The samples were collected using a novel sampling approach based on a volumetric absorptive microsampling technique that addresses the sample volume bias associated with the conventional blood cards. The OPLS-DA calibration model classified the subjects into their corresponding classes (FM and non-FM) with 98% accuracy, while the validation test gave good results with 0.88 AUC, 83% sensitivity, 85% specificity, and 84% accuracy. Amide bands and amino acids were important in describing the separation of FM from other related disorders. With the advances presented in the methodologies and subsequent analysis using Raman spectroscopy, an underlying biochemical difference among the diseases can be investigated, advancing the screening and treatment of the FM syndrome.

## Figures and Tables

**Figure 1 molecules-29-00413-f001:**
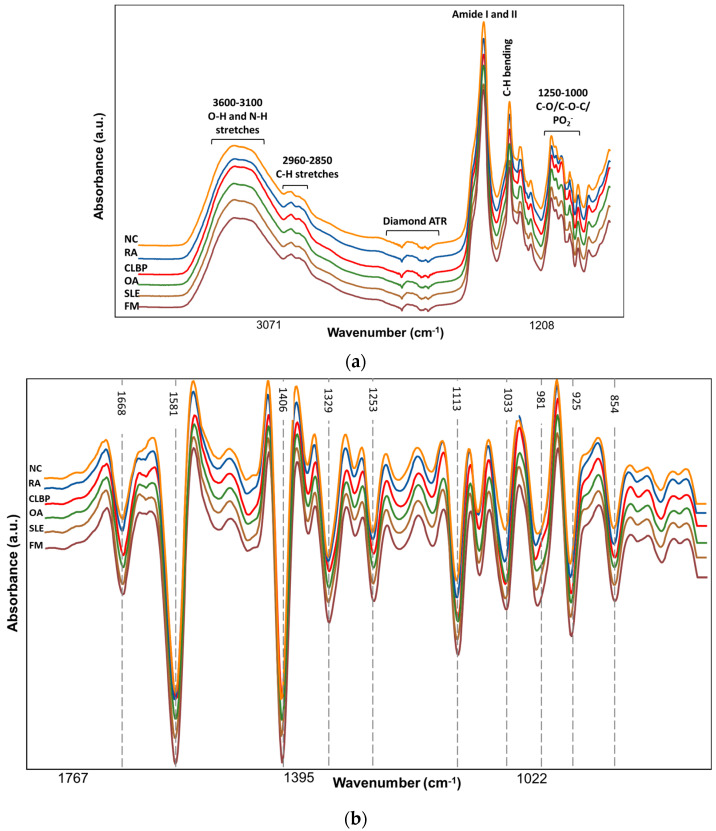
(**a**) Representation of spectral profile corresponding to the averaged data of individual diseases. Illustration of the major vibrational bonds and comparison of spectra among different groups. (**b**) Transformation (normalization and second derivative with 19 points) applied to the averaged spectra of FM, non-FM, and NC subjects.

**Figure 2 molecules-29-00413-f002:**
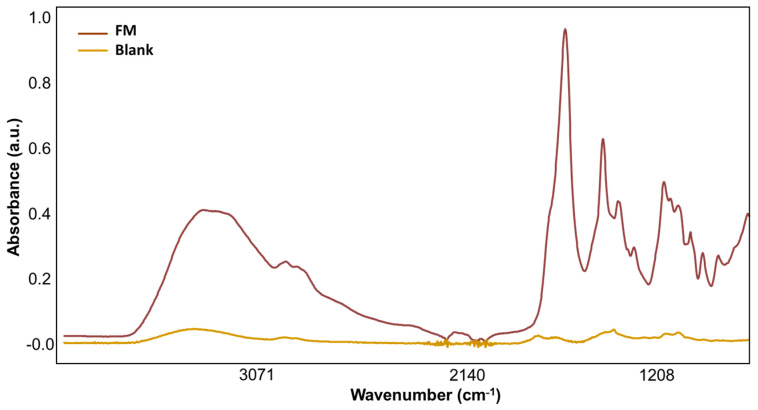
Representation of the spectral profile of an empty VAMS (red) and the sample absorbance (brown).

**Figure 3 molecules-29-00413-f003:**
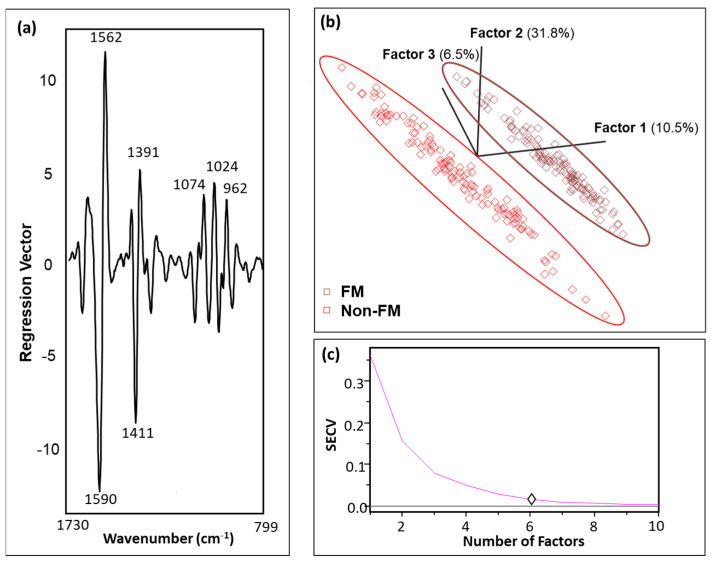
(**a**) Regression vector of the OPLS-DA model highlighting the important wavenumbers for identifying FM from the non-FM category. The 1590 and 1562 cm^−1^ region, corresponding to amide bands and amino acids, predominately governs the separation. (**b**) Score plot with the first three latent variables (LVs) of OPLS-DA regression calibration model obtained from the MIR spectral data. It depicts distinct separation of FM (brown) and non-FM (red) spectral attributes. (**c**) SECV versus number of factors plot illustrating an appropriate factor selection, 6.

**Figure 4 molecules-29-00413-f004:**
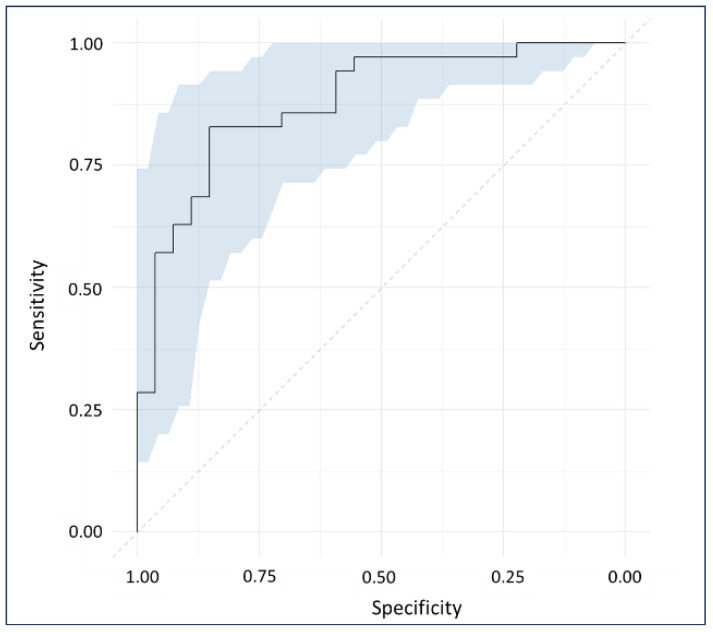
ROC plot showing 95% confidence interval of the AUC computed using DeLong method.

**Figure 5 molecules-29-00413-f005:**
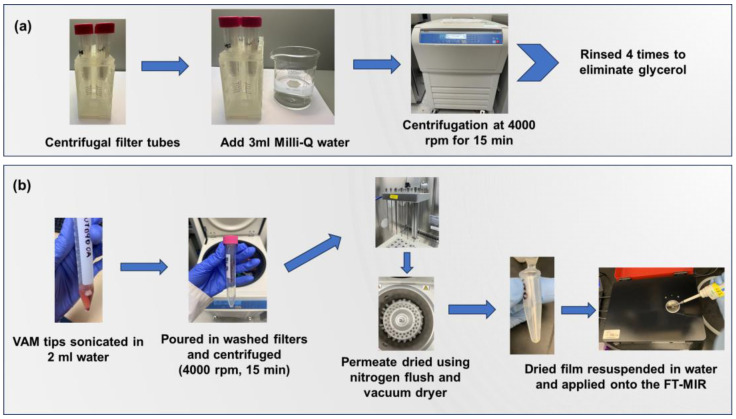
Sample preparation for extracting low-molecular serum fractions from VAMS tip, including (**a**) rinsing centrifugal filter membranes to remove glycerol coating and (**b**) using the washed membranes to filter serum fractions. The resulting permeate is dried to obtain a film which is resuspended in water for spectral analysis.

**Table 1 molecules-29-00413-t001:** Clinical characteristics of all subjects. Values expressed as mean ±/sd; N = number of subjects, age (range). FM: fibromyalgia; non-FM: rheumatoid arthritis (RA), systemic lupus erythematosus (SLE), osteoarthritis (OA), and chronic low back pain (CLBP). BMI: body mass index. CSI: Central Sensitization Inventory. SIQR: Symptom Impact Questionnaire Revised. FIQR: Fibromyalgia Impact Questionnaire Revised. McGill Pain Questionnaire (MPQ). BDI: Beck Depression Index.

	n	Age	(M/F) (%M/%F)	BMI	CSI	SIQR	FIQR	MPQ	BDI
FM	179	43.1 ± 13.4	(15/164) (8.3/91.7)	32.0 ± 9.2	58.1 ± 22.9		48.2 ± 26.0	90.5 ± 51.7	19.3 ± 11.8
Non-FM	158	50.5 ± 15.98	(36/122) (23/77)	28.2 ± 11.3	27.8 ± 21.9	33.5 ± 23.0		41.7 ± 49.6	9.5 ± 8.6
NC	13	42.3 ± 15.0	(5/8) (39/61)	25.5 ± 3.1	7.0 ± 10.5	1.7 ± 3.7		1.9 ± 3.8	1.1 ± 2.8

**Table 2 molecules-29-00413-t002:** Statistical analysis of FM and non-FM subjects. *p* values represent two-tailed comparisons between groups for FIQR/SIQR, BDI, MPQ, and CSI. Shapiro–Wilk normality test failed in each instance Mann–Whitney Rank Sum test passed. * Indicates statistically significant differences were found between groups.

	Age/Range	FIQR/SIQR	BDI	MPQ	CSI
FM	18–73	*	*	*	*
Non-FM	18–81	*p* < 0.05	*p* < 0.001	*p* < 0.001	*p* < 0.001

**Table 3 molecules-29-00413-t003:** Figures of merit of the OPLS-DA model, indicating standard error, coefficient of determination, sensitivity, specificity, and accuracy for differentiating FM class in calibration and validation datasets.

Figures of Merit	Calibration Set (*n* = 275)	Validation Set (*n* = 62)
SECV/SEP	0.02	0.02
R^2^	0.99	-
Sensitivity (%)	96	83
Specificity (%)	100	85
Accuracy (%)	98	84

**Table 4 molecules-29-00413-t004:** Table presenting a detailed distribution of samples corresponding to each disorder in the calibration and external validation dataset.

Dataset	FM	SLE	OA	RA	CLBP
Calibration	144	39	28	48	16
Validation	35	7	8	10	2
Total	179	158			

## Data Availability

The data presented in this study are available on request from the corresponding author. The data are not publicly available due to privacy concerns.

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
