# Peer review of "Portable Mid-Infrared Spectroscopy Combined with Chemometrics to Diagnose Fibromyalgia and Other Rheumatologic Syndromes Using Rapid Volumetric Absorptive Microsampling"

_molecules, 2024, doi:10.3390/molecules29020413_

Round 1

Reviewer 1 Report

Comments and Suggestions for Authors

Please clarify what is predicted /reported in Table 3.

Please provide more details about the second derivative. Please add the second derivative of the samples.

Please explain the criteria used to split the samples into cal and val.

It is not clear what is predicted or modelled. Please clarify.

Author Response

Thank you for reviewing our manuscript. We would like to thank the reviewers for their critical thinking and in-depth analysis. We have revised the manuscript accordingly and have highlighted those changes below.

Reviewer 1

  • Please clarify what is predicted /reported in Table 3.

We agree with the reviewer. To clarify what is reported in Table 3, we have included in the Results section the following sentences for better understanding of information shown in Table 3 (lines 238 to 246) “The OPLS-DA classification method incorporates the regression proficiencies of PLS approach where the response matrix is quantitative. The Y matrix included binary labels, 0 and 1 indicating non-FM and FM cases respectively. A leave-one-out cross validation approach was used to develop the calibration model in which each sample from the training set was temporarily left out and the remaining samples were used to build the model to predict the left-out sample. The cumulative Y residuals resulting from this internal validation was recorded as SECV. SEP is based on external validation and represents the residuals of the test set predictions and their corresponding known class labels. The SECV and SEP were similar with a low value of 0.02 (Table 3).” The SECV is reported based on the training set and SEP has been calculated based on the predicted data.

  • Please provide more details about the second derivative. Please add the second derivative of the samples.

We appreciate the reviewer suggestion. We have included in Figure 1b the second derivative of the FM, non-FM and NC subjects. We have also included further details about the second derivative in the Results section (lines 170 to 175): “The second derivative transformation was performed using a Savitzky-Golay polynomial filter [57]. In this method, consecutive subsets of the 19-point-sized windows were fitted by a second-order polynomial, followed by the application of a second derivative to the fitted polynomial curve [58]. Figure 1b compares the transformation (normalized and second derivatized with 19 points) applied to the averaged spectra of FM, non-FM and NC subjects.”

  • Please explain the criteria used to split the samples into cal and val.

A “Random” algorithm was used to select samples for calibration and validation data set. We have added the information in Materials and methods section (Lines 448 to 449) “The dataset was divided into training and validation sets using Random algorithm [28] (Table 4).”

  • It is not clear what is predicted or modelled. Please clarify.

We have added more information in the Results section to better explain what is predicted by our OPLS-DA algorithm (Lines 258 to 260) “Thus, our algorithm will correctly predict an unknown subject as FM or non-FM with an 84% accuracy rate.” We are modelling the response variable (Y) that includes 2 classes (FM and non-FM). We have also included the sentence (Lines 239-240)The Y matrix included binary labels, 0 and 1 indicating non-FM and FM cases respectively.”

Reviewer 2 Report

Comments and Suggestions for Authors

A method for diagnosing fibromyalgia (FM) was proposed in this paper. The method uses prepared blood samples with volumetric absorptive micro-sampling (VAMS), portable FT-IR and chemometric algorithm (OPLS-DA) to discriminate the FM and non-FM samples. The method may be practicable and the result seems acceptable. Important wavenumbers were identified through the regression vector of the OPLS-DA model, which may be helpful for understanding the difference between FM and non-FM samples. The paper may be publishable after minor revisions.

1. The author indicated on page 3, lines 119-120, that “The objective of the current study was to develop a biomarker-based diagnostic approach using the characteristic infrared signatures of blood samples.” Generally, this is impossible, because PLS-DA is based on the full information of the selected spectral range, although the regression vector is helpful.

2. It is confusing that, in Figure 3(b), the 3D projection was plotted in a 2D space. On the other hand, the number in the brackets should be the explained variance. Why the number for Factor 1 is smaller than that for Factor 2? 

3. For the results in Table 3 that should be obtained with PLS-DA, how were SECV/SEP and R2 calculated? If R2 for calibration set can be calculated, the value for the validation set should also be calculated. On the other hand, all the three parameters of sensitivity, specificity and accuracy for the calibration set are better than that of the  validation set. Was the model overfitted?

Author Response

Thank you for reviewing our manuscript. We would like to thank the reviewers for their critical thinking and in-depth analysis. We have revised the manuscript accordingly and have highlighted those changes below.

Reviewer 2

A method for diagnosing fibromyalgia (FM) was proposed in this paper. The method uses prepared blood samples with volumetric absorptive micro-sampling (VAMS), portable FT-IR and chemometric algorithm (OPLS-DA) to discriminate the FM and non-FM samples. The method may be practicable, and the result seems acceptable. Important wavenumbers were identified through the regression vector of the OPLS-DA model, which may be helpful for understanding the difference between FM and non-FM samples. The paper may be publishable after minor revisions.

  • The author indicated on page 3, lines 119-120, that “The objective of the current study was to develop a biomarker-based diagnostic approach using the characteristic infrared signatures of blood samples.” Generally, this is impossible, because PLS-DA is based on the full information of the selected spectral range, although the regression vector is helpful.

We thank the reviewer for this suggestion. We have removed “biomarker-based” (Line 120) from the description of our research objective.

  • It is confusing that, in Figure 3(b), the 3D projection was plotted in a 2D space. On the other hand, the number in the brackets should be the explained variance. Why the number for Factor 1 is smaller than that for Factor 2?

We agree with the reviewer. The score plot is a 2D representation of the first three latent variables (LVs) of OPLS-DA regression calibration model obtained from the MIR spectral data. We have changed the Figure 3b caption to better explain the information showed in this plot “Score plot with the first three latent variables (LVs) of OPLS-DA regression calibration model obtained from the MIR spectral data”.  Regarding the increase in the variance, we have added a paragraph in the Results section (Lines 216 to 223) “The explained variation of the X-matrix by the factors (Figure 3b) did not follow a gradual decrease, as seen in principal component analysis (PCA). This deviation is explained by the fact that OPLS-DA emphasizes the relationship between the predictive variables in X and the Y response [59,60]. The OSC filter selectively removes the systematic X variations that are not response-predictive, focusing on factors that maximize the discrimination among the classes [59]. This contrasts with PCA, which captures and maximizes the overall variation within the X matrix [60]. Nevertheless, with inclusion of more factors, the associated SECV decreased (Figure 3c).” Additionally, we have included SECV versus number of factors plot as Figure 3c with description: “(c) SECV versus number of factors plot illustrating an appropriate factor selection, 6.”

  • For the results in Table 3 that should be obtained with PLS-DA, how were SECV/SEP and R2 calculated? If R2 for calibration set can be calculated, the value for the validation set should also be calculated. On the other hand, all the three parameters of sensitivity, specificity and accuracy for the calibration set are better than that of the validation set. Was the model overfitted?

The SECV, SEP and calibration R2 was calculated by the Pirouette software. Correlation coefficient measures the linear relationship between the predicted values and the known values. Its significance is more prominent in the regression analysis where the focus is on predicting the quantity of the Y variable. Although, it may not be the most important metric in classification algorithm which emphasizes class separation rather than establishing a linear relationship.

We have included more information in the Results section to better explain the performance of our OPLS-DA algorithm (Lines 253 to 257) “OPLS-DA has powerful modeling features that facilitate separation based on experimental classes in the calibration model, yielding positive results [59]. An external validation with an unseen dataset is required to validate the statistical significance of the separation and draw conclusions [59].” We are aware that considering too many factors increases the probability of modelling the noise leading to overfitting. For our OPLS-DA model, we selected an optimal number of factor (6) using the standard error versus number of factors plot that levels between underfitting and overfitting. We have included the plot in the results section (Figure 3c) and explained it in the paragraph (Line 209) “Six factors were selected (Figure 3c) that cumulatively described 81.4% of the variation.”